# Impact of Entrepreneurship on Innovation Performance of Chinese SMEs: Focusing on the Mediating Effect of Enterprise Dynamic Capability and Organizational Innovation Environment

**Fang Cui [1] and Jae-hoon Song [2,\*]**

[1] School of Economics and Management, Taishan University, Tai'an 271000, China
[2] Department of Social Economics and Management, Graduate School, Woosuk University, Wanju-gun 55338, Korea
\* Correspondence: jhsong@woosuk.ac.kr

**Abstract:** The primary objective of this study is to examine the relationship of entrepreneurship, enterprise innovation performance, enterprise dynamic ability, and organizational innovation environment of small and medium enterprises in China. Based on a thorough literature review, this study constructed a research model of "entrepreneurship, enterprise dynamic ability, organizational innovation environment, and enterprise innovation performance". To meet this purpose, a survey was conducted by collecting data using a random sampling method targeting five cities in Shandong and Henan, where the proportion of small and medium-sized enterprises (SMEs) in China is high. Additionally, SPSS 26.0 and AMOS 24.0 were used to carry out correlation analyses, structural equation path analyses, and intermediary effect tests on 426 valid questionnaires. There are three major findings. First, entrepreneurship positively affects the dynamic ability of enterprises and the organizational innovation environment and enterprise innovation performance. Second, enterprise dynamic capability and organizational innovation environment positively affect enterprise innovation performance. Third, enterprise dynamic capability and organizational innovation environment play a partial mediating role on the influence path between entrepreneurship and innovation performance. The findings of this paper provide further evidence for the positive impact of entrepreneurship on innovation performance and the relationship between variables. Suggestions are provided for SMEs to improve their innovation performance. To stimulate entrepreneurship of SMEs, more attention needs to be given to cultivate their dynamic capabilities, create an environment of organizational innovation for enterprises, and help them maintain a competitive advantage in the ever-changing market environment.

**Keywords:** entrepreneurship; innovation performance; enterprise dynamic capability; organizational innovation environment; small and medium-sized enterprises



## 1. Introduction

At present, China's economy is in a transition period and is also in a recovery period after a major public health crisis. The market environment is extremely complex and uncertain. Such a market environment provides both opportunities but also poses challenges for SMEs. Most managers of small and medium-sized enterprises have a substantial profit orientation, only pay attention to the demand for profits, and blindly expand to the fields that make money quickly. Some managers have thoughts and behaviors of worrying about gains and losses, being content with the status quo, lacking responsibility for building national brands and achieving national undertakings, and even having integrity problems such as failing to perform contracts and making false advertisements. Therefore, the lack of entrepreneurship is an essential factor that hinders the high-quality development of

and improves their innovation performance. To promote the high-quality development SMEs of small and medium-sized enterprises, it is urgent to clarify the new connotation of entrepreneurship and carry forward good entrepreneurship. Hu [1] pointed out that enterprise managers with entrepreneurship need to stimulate enterprise vitality, promote enterprise innovation, and seek new development opportunities. Enterprise dynamic capabilities may help enterprises innovate, integrate, and reconstruct, and gain competitive advantages in a dynamic competitive environment. Schumpeter [2] pointed out that innovation promotes economic growth and development, and entrepreneurs are the main body of innovation. The new era of economic development needs more entrepreneurs with entrepreneurial spirit. Entrepreneurship enhances the competitiveness of enterprises and serves as a source of the core competitiveness of enterprises. Small and medium-sized enterprises play a vital role in the development of China's economy. Therefore, the improvement of the innovation performance of SMEs is essential to the high-quality economic development.

Knight Frank [3] put forward the concept of entrepreneurship, and he regarded entrepreneurship as the talents and abilities of entrepreneurs, including the innovative spirit of blazing a trail and the spirit of taking risks to create creative activities. Waterman et al. [4] believed that entrepreneurship has a positive impact on firm performance and may bring dynamism to firms. Peter Drucker [5] believed that the world economy transformation calls for more entrepreneurial in terms of innovation development. Strengthening entrepreneurship and a dedication to innovation can create enterprise vitality. Chen et al. [6] found that environmental uncertainty has a positive impact on firm performance through entrepreneurship and organizational learning. Liu's [7] research showed that entrepreneurship is conducive to enhancing corporate innovation performance.

The research of enterprise dynamic capability originated from the core capability theory in the research field of strategic management. The ever-changing external environment urges enterprises to constantly update their core capabilities of innovation to maintain a sustainable competitive advantage. The concept of dynamic capabilities was first proposed by Teece et al. [8]. They believed that to timely respond to the changing environment, enterprises need the capability to continuously build, reshape, configure, and reconfigure technologies and resources inside and outside the enterprise. Helfat [9] defined dynamic capabilities as competencies that enable enterprises to respond to environmental changes by producing new products or reconfiguring production processes. Eisenhardt and Martin [10] stated that enterprise integration, acquisition, reconstruction, and resource allocation is the outcome of dynamic capabilities in response to the changes from the external environment. Helfat [11] emphasized the role of enterprise managers in his subsequent research and stated that enterprise dynamic capabilities are managers' key abilities to purposefully develop, expand, or change their resources. Jiao [12] found through empirical evidence that the entrepreneurship of managers is an essential factor to enterprise dynamic capabilities. Shi et al. [13] believed that dynamic capability is not only a comprehensive application capability, but also the intrinsic spiritual energy in enterprise development. It can be seen from the above discussion that there is no consensus on the definition of the dynamic capabilities. In this study, enterprise dynamic capabilities are defined as the ability of enterprises to maintain a sustainable competitive advantage in the changing environment.

Innovation environment is a component of organizational environment. Payne and Wall [14] proposed the concept of innovation environment. They explained that organizational innovation environment is the work environment related to organizational innovation subjectively perceived by individuals in the enterprise to the innovative elements in the surrounding environment. Schneider et al. [15] put forward the concept of innovation environment based on organizational environment. They defined innovative environment as employees' subjective perception of the innovative elements in their working environment. Wu and You [16] found that the innovation environment has a direct impact on the innovation performance of enterprises. Wang and Chang [17] believed that

innovation environment is a special organizational characteristic that exists within the organization and can be directly or indirectly perceived by members to support innovation at the overall level of the organization. The innovative environment of an organization can influence its employees' innovative attitudes and behaviors.

Enterprise innovation performance can be seen as the efficiency of the enterprise innovation process, the effect of output, and its contribution to the business success of the enterprise [18]. Ari [19] defined enterprise innovation performance as the result of promoting the improvement of enterprise benefits through innovation activities, including technological innovation process and product innovation process. Alegre et al. [20] summarized the evaluation indicators of innovation performance and summarized that innovation performance represents the performance of enterprises to obtain resources and allocate resources to improve the efficiency of resource allocation and obtain higher economic results.

The difference of this study is that the past research on SMEs in China focused on securing external competitiveness such as marketing strategy, and there was insufficient research on securing internal competitiveness of SMEs. To clarify this, the goal of this research is to know the innovation performance of Chinese firms through entrepreneurship to increase the competitiveness of Chinese SMEs. This study is composed of five parts, and the contents of each part are arranged as follows: the first part is the introduction of this study, the second part is the literature review and research assumptions, the third part is the research design, the fourth part is the data analysis results, and the fifth part is the research conclusions and implications.

The research contributions of this study are divided into theoretical contributions and practical contributions.

In terms of theoretical contributions, the relationship between entrepreneurship and enterprise innovation performance has always been the focus of academic attention. Joseph Alois Schumpeter, Peter F. Drucker, and many other scholars have made in-depth discussions on this issue from multiple perspectives. This paper studies entrepreneurship, enterprise innovation performance, enterprise dynamic capability, and organizational innovation environments in a model based on the new requirements of China's economy for SMEs. To explore the relationship between the four, it is important to test the mediating role of enterprise dynamic capabilities and organizational innovation environment between entrepreneurship and innovation performance and analyze the mechanisms and the reasons for the influence, therefore enriching the research on the effect of entrepreneurship on enterprise innovation performance.

Practical contribution aspects, through theoretical and empirical research, put forward the corresponding management suggestions, providing a new theoretical basis for the innovation and development of SMEs, which is conducive to the improvement of enterprise innovation performance. For entrepreneurs of SMEs, to give full play to their entrepreneurial spirit in daily management activities, entrepreneurs themselves should persist in continuous innovation and have the courage to challenge. They should create a positive environment for organizational innovation, influence employees with entrepreneurial spirit, actively grasp the market dynamics and industry development trends, use internal and external resources to absorb knowledge, integrate and develop various resources, make use of them so as to improve the dynamic capabilities of the enterprise, and finally, promote the improvement of enterprise innovation performance.

## 2. Literature Review and Research Hypothesis

### 2.1. The Impact of Entrepreneurship on the Dynamic Capabilities

Zahra [21] found that entrepreneurship activities help small and medium-sized enterprises overcome the crisis as they grow, which is beneficial to enhance their dynamic capabilities and build competitive advantages. Han et al. [22] conducted in-depth interviews with 368 enterprises and used structural equation modeling to empirically test that entrepreneurship positively and significantly affects the dynamic capabilities of enterprises.

The research of Ma et al. [23] also revealed that entrepreneurship positively affects the formation of dynamic capabilities of enterprises. This study regards entrepreneurship as a comprehensive capability, with entrepreneurship managers able to make adjustments in business strategies and give responses to the changes from the external environment. Entrepreneurship can also help managers integrate and reconstruct resources for the organization when necessary. In SMEs, the entrepreneurship may enhance the dynamic capabilities of the enterprise. Thus, this study proposes the following hypothesis.

**Hypothesis 1.** *Entrepreneurship has a positive impact on enterprise dynamic capabilities.*

### 2.2. The Impact of Entrepreneurship on Organizational Innovation Environment

The research of Scott and Bruce [24] revealed that the entrepreneur's support for innovation and the creation of an innovative environment might give employees a great psychological motivation, which can ultimately affect innovation performance. Drucker [25] defined entrepreneurship as a heterogeneous resource of an enterprise and an important driver of enterprise innovation. Yuan's [26] research indicated that entrepreneurship and organizational innovation environment have a significant positive correlation. Fan [27] found that entrepreneurship has a significant positive impact on organizational innovation environment. Entrepreneurship may help enterprises to seek business opportunities, create an environment of organizational innovation, and promote the innovation and growth of enterprises. Thus, this study proposes the following hypothesis.

**Hypothesis 2.** *Entrepreneurship has a positive impact on organizational innovation environment.*

### 2.3. The Impact of Enterprise Dynamic Capabilities on Enterprise Innovation Performance

Zollo [28] found that enterprises with dynamic capabilities may create opportunities for resource integration and reconstruction, which can help enterprises obtain excess profits. Ettlie [29] took automobile enterprises as the research sample and found that enterprise dynamic capabilities play an important role in the development of new products and the promotion of new products. Jiang [30] believed the enterprises' dynamic capabilities are an invisible ability and exist in enterprises among leaders and employees. This invisible ability might affect the innovative behavior of employees and affect corporate innovation performance. Shi [31] pointed out that there is a positive correlation between the dynamic capabilities and the innovation performance in enterprises. Thus, this study proposes the following hypothesis.

**Hypothesis 3.** *Enterprise dynamic capabilities have a positive impact on enterprise innovation performance.*

### 2.4. The Influence of Organizational Innovation Environment on Enterprise Innovation Performance

Cooper [32] claimed that an organizational innovation environment would bring about enterprises' core competitiveness, and unique competitive advantages are beneficial for the innovation performance, so innovation environment is a key factor to the formation of innovation performance. Mumford et al. [33] studied the relationship between innovation environment and innovation performance. He found that the average correlation coefficient between innovation environment and innovation performance was 0.35, and there was a significant correlation between the two variables. Frei [34] studied the influence of organizational innovation environment on innovation performance from the perspective of knowledge diffusion. He found out that innovation environment makes employees form a positive knowledge sharing attitude, which may significantly improve employees' learning behaviors and affect innovation performance. Both Zhang [35] and Zhao [36] found that there is a significant positive correlation between organizational innovation environment and innovation performance. Therefore, this study proposes the following hypothesis.

**Hypothesis 4.** *Organizational innovation environment has a positive impact on enterprises innovation performance.*

*2.5. The Impact of Entrepreneurship on Enterprise Innovation Performance*

Schumpeter et al. [37] pointed out that innovation requires the entrepreneur's ability to commercialize innovative ideas. Even if entrepreneurs have the ability to identify opportunities, if they do not utilize opportunities, they cannot create competitive advantages. Only by transforming potential opportunities into operational strategies in innovation activities can they turn ideas into heterogeneous resources and make them more effective, resulting in businesses gaining new competitive advantages. Shane et al. [38] pointed out that entrepreneurship can help enterprises identify opportunities to enable enterprises to obtain sustainable competitive advantages, thereby improving the innovation performance of enterprises. Banda [39] studied data from 22 OECD countries and pointed out that entrepreneurship might positively promote economic growth. Chen et al. [40] designed an entrepreneurship scale based on research on entrepreneurship and concluded that entrepreneurship can positively promote enterprise innovation performance through interviews and questionnaires (Supplementary Materials File S1). Wu [41] believes that entrepreneurs with a strong sense of innovation have a strong initiative to put them into practice or motivate their employees to jointly realize innovative behaviors and promote innovative performance. Brańka [42] investigated SMEs and found that strong entrepreneurship affects the innovation performance of enterprises. Liu [43] found that managers with good entrepreneurship may provide effective support for corporate innovation performance. Based on these findings, this study proposes the following hypothesis.

**Hypothesis 5.** *Entrepreneurship has a meaningful positive impact on enterprises innovation performance.*

*2.6. The Mediating Role of Dynamic Capability and Organizational Innovation Environment*

Wei's [44] research revealed a mediating effect between enterprise dynamic capability and enterprise innovation performance. Ren [45] also proved the mediating effect of enterprise dynamic capability between entrepreneurship and enterprise innovation performance. The mediating role of enterprise dynamic capability can be achieved through learning absorptive capacity, resource integration capacity, and transformation and reorganization capacity. Hu's [1] research shows that enterprise dynamic capability plays a mediating role between entrepreneurship and innovation performance, but dynamic capability plays a partial mediating role in this influence path. Entrepreneurship will affect enterprise dynamic ability. With the increase of entrepreneurship, the dynamic capabilities of enterprises will also improve. Strong dynamic capabilities may bring more innovative behaviors, and the innovation performance of enterprises may also be improved. Based on this, this study proposes the following hypothesis.

**Hypothesis 6.** *Dynamic capabilities play a significant mediating role in the relationship between entrepreneurship and enterprise innovation performance.*

Both Schumpeter's [37] and Drucker's [5] innovation theories provided evidence for the mediating role of firm innovation environment between entrepreneurship and firm innovation performance. Xia [46] found that entrepreneurship improves employee innovation capability through organizational internal innovation environment, thereby improving corporate innovation performance and promoting corporate innovation growth. Therefore, entrepreneurship can maintain the sustainable competitive advantage and growth of enterprises by creating an environment of organizational innovation. That is, the environment of organizational innovation serves as an intermediate for entrepreneurship to promote enterprise growth. The research of Li et al. [47] shows that strengthening motivation and working capabilities and a good corporate innovation environment are conducive to

enhancing entrepreneurship, thereby promoting the generation of corporate innovation performance. Enterprise innovation environment is a medium for entrepreneurship to be transmitted to enterprise growth. Based on the research of Li and other scholars, this study proposes the following hypothesis.

**Hypothesis 7.** *Organizational innovation environment plays a mediating role in the relationship between entrepreneurship and enterprise innovation performance.*

According to the seven hypotheses mentioned above, the specific theoretical model proposed in this study is shown in Figure 1.

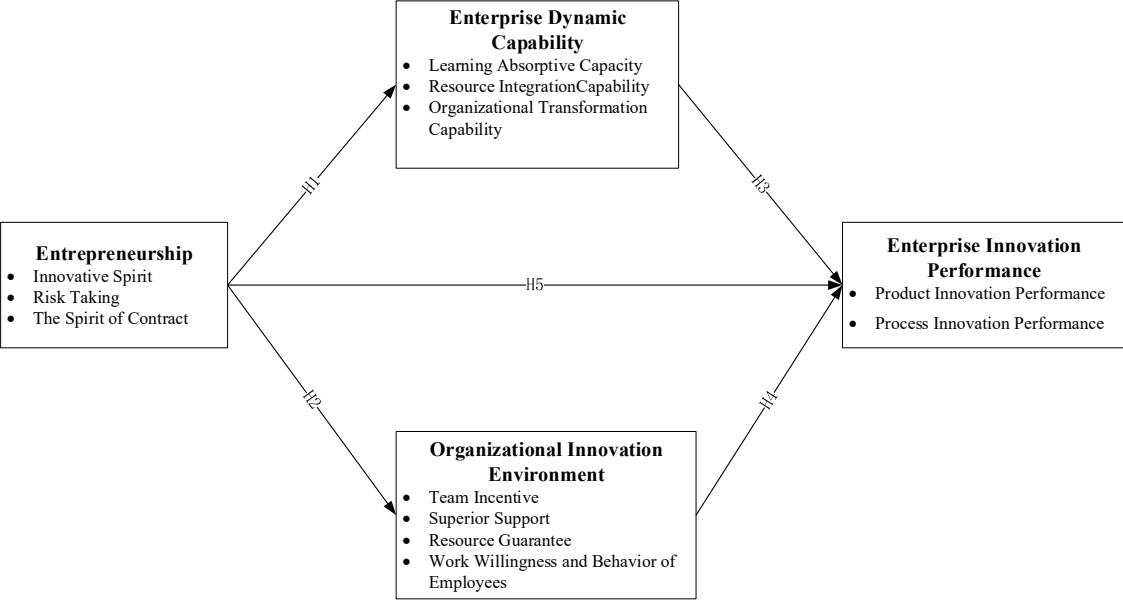

**Figure 1.** Conceptual model.

## 3. Research Design

### 3.1. Research Sample and Data Collection

This study's main objective is to explore entrepreneurship's impact on innovation performance. The research participants are mainly managers in small and medium-sized enterprises. The types of enterprises are mainly concentrated in the retail, manufacturing, and real estate industries. In terms of enterprise scale, the definition of SMEs is mainly based on the *Notice on Printing and Distributing the Provisions on the Classification Standards for Small and Medium-sized Enterprises* (2011) issued by relevant state agencies. There are two main reasons. First, SMEs are the backbone of China's economy and social development and have high research value; second, in SMEs, the role of managers' entrepreneurship is more obvious, which has a significant impact on the dynamic capabilities of enterprises and organizational innovation. The effect of innovation environment creation and enterprise innovation performance may also be more obvious.

This study collected data by issuing questionnaires. Using the Likert five-point scale, respondents were asked to score according to their true situation and the consistency of the items. In the questionnaire setting, basic information such as the establishment year, scale, and type of the enterprise are added. The selected scales are mature scales that predecessors have used many times. The managers of small and medium-sized enterprises were investigated by simple random sampling. This study started on 15 April 2022 and ended on 15 May 2022. In five cities, including Zhengzhou City in Henan Province and Jinan City in Shandong Province, a total of 500 questionnaires were distributed by on/off-line and there were 426 responses, with a response rate of 85.2%.

*3.2. Measurement Tools*

3.2.1. Measurement of Entrepreneurship

Entrepreneurship is the driving force for an enterprise to overcome internal and external difficulties and build a long-term business, and it is the key to successfully building an enterprise's developing power. Entrepreneurship is manifested as innovative spirit, risk-taking spirit, and contract spirit. It is reflected in the enterprise's never-ending pursuit of value. Based on the previous definition of entrepreneurship, this paper focuses on innovative spirit, risk-taking, and contract spirit.

Entrepreneurship is measured from the three dimensions of innovative spirit, risk-taking, and contract spirit. First, in terms of innovative spirit, Covin and Slevin [48] measured entrepreneurship from the three dimensions of innovation, risk-taking, and pioneering. The internal consistency of the scale was 0.937, and the consistency of the three factors of innovation, risk-taking, and pioneering was 0.857, 0.853, and 0.842, respectively, which indicates good construct validity. The scale was widely adopted by scholars at home and abroad. This paper adopts the scale developed by Covin and Slevin [48] to measure entrepreneurial innovation. Second, in risk-taking spirit aspects, risk-taking is an entrepreneur's attitude toward risk and uncertainty. Based on the scale developed by Covin and Slevin [48], this paper measures the entrepreneur's risk-taking spirit from the aspects of the entrepreneur's attitude towards uncertainty, risk-taking, and the degree of risk-taking. Third, in the spirit of the contract aspects, based on the literature analyses of social contract, psychological contract, and contract performance, this study adopted the scales developed by Caroll [49], Lu et al. [50], Shi et al. [51], and Yan Ling et al. [52]. The spirit of the contract is measured from the aspects of the entrepreneur's performance of the agent contract, psychological contract, commercial contract, and social contract, as well as integrity and legal awareness.

3.2.2. Measurement of Enterprise Dynamic Capabilities

In this study, enterprise dynamic capability is the mediating variable. From the existing literature, due to the different perspectives of researchers, the division of various dimensions of enterprise dynamic capability is not the same. Most scholars start from the perspective of resource integration, and utilization dynamic capabilities are divided into learning absorptive capabilities, resource integration capabilities, and transformation capabilities [10,53–55]. The dynamic capabilities of enterprises are measured in three dimensions: learning absorptive capacity, resource integration capacity, and organizational transformation capacity.

3.2.3. Measurement of Organizational Innovation Environment

Some scholars developed measurement scales to measure organizational innovation environment. Siegel [56] designed the SSSI scale according to the dimension of organizational innovation environment. The scale is used to detect the innovation environment in the European education field, with a total of 61 items. The verification shows that this scale has certain credibility. Amabile et al. [57] developed the WEI scale when they studied the influencing factors of organizational innovation environment among corporate R&D and sales personnel, and revised the scale in 1996, which has also been used by subsequent scholars many times. The reliability and validity of the KEYS scale has also been verified. Sun et al. [58] and Lian et al. [59] revised or compiled an organizational innovation environment measurement scale suitable for Chinese companies based on the reality of Chinese corporate culture based on the above-mentioned organizational innovation environment measurement scale, combined with the division dimensions of organizational innovation environment in knowledge-intensive enterprises in this paper. Organizational innovation environment is measured in three dimensions: team incentives, superior support, resource guarantee, employee work willingness, and behavior. The relevant scales are slightly modified to form the organizational innovation environment measurement scale for this study.

### 3.2.4. Measurement of Innovation Performance

The measurement of enterprise innovation performance adopts the two basic dimensions: product innovation performance and process innovation performance. Altogether, the two dimensions adopt seven indicators proposed by Prajgo and Sohal [60]. Product innovation performance include four indicators. First, product development ability. Second, the commercialization speed of products. Third, the ability of new products to maintain and improve market profitability. Fourth, market share of new products. Process innovation performance include three indicators: First, the speed of work tasks, decision-making, and information system innovation, as well as product innovation and process innovation. Second, product quality, process quality, process flexibility, and the ability to reduce production costs. Third, in the process of innovation, accept the speed of process innovation. The respondents were asked to compare the enterprise with other enterprises in the industry and select the description content that they most agree with or think is the most appropriate according to the actual situation of the company in the past three years and score.

### 3.3. Statistical Analysis

This paper uses SPSS26.0 and AMOS24.0 software for data processing and analysis. Firstly, the basic situation of the sample is investigated through descriptive analysis. Secondly, the internal consistency of the data, and the discriminant validity between variables. Finally, the internal relationship between variables is examined through correlation analysis, structural equation path analysis, and mediation analysis.

## 4. Data Analysis Results

### 4.1. Demographic Information

The demographic characteristics of this survey mainly include gender, age, tenure, position level, company establishment, company size, and enterprise type. The gender distribution of the sample is relatively balanced, with men and women accounting for 52.6% and 47.4%, respectively. The age is mainly middle-aged and elderly, and the cumulative proportion of 36–45 years old, 46–55 years old, and over 55 years old is 67.6%. The tenure of office is basically more than 4 years, accounting for 77.7%, and only 22.3% are under 4 years. In terms of job levels, senior managers, middle managers, and grassroots managers each account for about 1/3. Both start-ups and mature companies are covered, with the highest proportion of public companies with 6–10 years of establishment at 23.2%. The size of the company is mainly 51–100 people, accounting for 22.3%. The types of enterprises are mainly concentrated in the retail, manufacturing, and real estate industries. On the whole, the sample covers all levels and is representative to a definite extent.

In this study, the skewness and kurtosis of the measurement items of each variable meet the conditions of normal distribution.

### 4.2. Reliability and Confirmatory Factor Analysis

Since all the mature scales are used in this paper, the combined reliability and discriminant validity are also tested by confirmatory factor analysis, in addition to the reliability $\alpha$ test.

It can be seen from the data in Table 1 that the reliability $\alpha$ of the latent variables entrepreneurship, enterprise dynamic capability, organizational innovation environment, and enterprise innovation performance are 0.948, 0.934, 0.931, and 0.884, respectively, all of which are above 0.8, which meet the statistical requirements and have good reliability.

It can be seen from the data in Table 2, in the confirmatory factor analysis, that the model fits well: $\chi^2/df$ = 0.882, which is between 1 and 3; GFI = 0.984, AGFI = 0.974, all greater than 0.8; RMSEA = 0.000, RMR = 0.015, all less than the decisive value of 0.08; IFI = 1.003, TLI = 1.004, CFI = 1.000, the indicators are all greater than the basic requirements of 0.9. All factor loadings are between 0.722–0.806. The larger the factor loading value, the more effectively the measurement item can reflect the dimension content to be measured. The standardized factor loading of latent variables is greater than 0.5, and

the factor loading of each measurement item is greater than 0.5. All are significant at the 0.001 level, which indicates that each measurement item can reflect its dimension. The compositional reliability of latent variables entrepreneurship, enterprise dynamic capability, organizational innovation environment, and enterprise innovation performance are 0.798, 0.816, 0.835, and 0.699, respectively, all of which are greater than the basic threshold of 0.7. It can be seen that the compositional reliability of the overall scale is good. The average variance extraction amounts were 0.569, 0.597, 0.559, and 0.537, which are all greater than the basic threshold of 0.5, indicating that the overall scale had good convergent validity.

**Table 1.** Reliability analysis results of the scales.

| Scales | Variables | Cronbach's Alpha | |
|---|---|---|---|
| Entrepreneurship | Innovative spirit | 0.919 | |
| | Risk-taking | 0.908 | 0.948 |
| | Contract spirit | 0.931 | |
| Enterprise Dynamic Capability | Learning absorptive capacity | 0.918 | |
| | Organizational transformation capability | 0.922 | 0.934 |
| | Resource integration capability | 0.841 | |
| Organizational innovation environment | Team incentive | 0.899 | |
| | Superior support | 0.866 | |
| | Resource guarantee | 0.855 | 0.931 |
| | Work willingness and behavior of employees | 0.881 | |
| Enterprise innovation performance | Product innovation performance | 0.894 | |
| | Process innovation performance | 0.847 | 0.884 |

**Table 2.** CFA result.

| Question Items | | Variables | SE | C.R. | $p$ | Standardized Load | CR | AVE |
|---|---|---|---|---|---|---|---|---|
| Innovative spirit | ← | Entrepreneurship | | | | 0.737 | | |
| Risk-taking | ← | Entrepreneurship | 0.069 | 13.837 | *** | 0.797 | 0.798 | 0.569 |
| Contract spirit | ← | Entrepreneurship | 0.070 | 13.128 | *** | 0.727 | | |
| Learning absorptive capacity | ← | Enterprise Dynamic Capability | | | | 0.770 | | |
| Organizational transformation capability | ← | Enterprise Dynamic Capability | 0.080 | 14.527 | *** | 0.792 | 0.816 | 0.597 |
| Resource integration capability | ← | Enterprise Dynamic Capability | 0.057 | 14.139 | *** | 0.755 | | |
| Team incentive | ← | Organizational innovation environment | | | | 0.722 | | |
| Superior support | ← | Organizational innovation environment | 0.075 | 14.798 | *** | 0.806 | | |
| Resource guarantee | ← | Organizational innovation environment | 0.065 | 13.761 | *** | 0.738 | 0.835 | 0.559 |
| Work willingness and behavior of employees | ← | Organizational innovation environment | 0.080 | 13.479 | *** | 0.721 | | |
| Product innovation performance | ← | Enterprise innovation performance | | | | 0.751 | | |
| Process innovation performance | ← | Enterprise innovation performance | 0.066 | 11.963 | *** | 0.714 | 0.699 | 0.537 |

*** $p < 0.001$, $\chi^2/\mathrm{df} = 0.882$, GFI = 0.984, AGFI = 0.974, RMR = 0.015, RMSEA = 0.000, IFI = 1.003, TLI = 1.004, CFI = 1.000.

### 4.3. Correlation and Discriminant Validity

This study uses the Pearson correlation coefficient to measure the correlation between variables and the AVE method to evaluate the discriminant validity. The results are shown in Table 3.

**Table 3.** Inter construct correlation table with $\sqrt{}$AVE scores.

|  | 1 | 2 | 3 | 4 |
|---|---|---|---|---|
| Entrepreneurship | **0.754** | | | |
| Enterprise Dynamic Capability | 0.333 ** | **0.766** | | |
| Organizational innovation environment | 0.449 ** | 0.357 ** | **0.747** | |
| Enterprise innovation performance | 0.489 ** | 0.459 ** | 0.518 ** | **0.733** |
| Mean | 3.734 | 3.699 | 3.715 | 3.760 |
| Standard deviation | 0.685 | 0.752 | 0.698 | 0.749 |

** $p < 0.05$; Correlation is significant at the 0.01 level (2-tailed). Note: 1–4 represent entrepreneurship, enterprise dynamic capability, organizational innovation environment, and enterprise innovation performance.

It can be seen from the above table that the correlation coefficients between entrepreneurship and enterprise dynamic ability, organizational innovation environment, and enterprise innovation performance are 0.333, 0.449, and 0.489, respectively, and the $p$ values reached a significant level of 0.05, indicating that entrepreneurship and enterprise dynamic ability, organizational innovation environment, and corporate innovation performance have a significant positive correlation; the correlation coefficients of corporate dynamic capability, organizational innovation environment, and corporate innovation performance are 0.459 and 0.518, respectively, and the $p$ values reached a significant level of 0.05, indicating that the enterprise has significant correlations with dynamic capabilities, organizational innovation environment, and corporate innovation performance.

The value range of the correlation coefficient between variables is 0.333~0.518, and the value scope of the AVE squared root values are between 0.733~0.766. It can be seen that the unconditional value of the correlation coefficient between variables is smaller than the variable AVE squared root value, indicating that the research variables in this paper have good discriminant validity.

*4.4. Structural Regression Model*

The above confirmatory factor analysis shows that the measurement model fits well. Therefore, the path analysis is carried out. The path coefficients and significance levels of this study are proved in Table 4.

**Table 4.** Path coefficients for structural model.

| Path | Standardized Coefficient | S.E. | C.R. | $p$ |
|---|---|---|---|---|
| Entrepreneurship → enterprise dynamic capability | 0.453 | 0.067 | 7.241 | *** |
| Entrepreneurship → organizational innovation environment | 0.571 | 0.064 | 8.782 | *** |
| Entrepreneurship → enterprise innovation performance | 0.322 | 0.062 | 5.204 | *** |
| enterprise dynamic capability→ enterprise innovation performance | 0.367 | 0.075 | 5.298 | *** |
| organizational innovation environment → enterprise innovation performance | 0.330 | 0.083 | 4.254 | *** |

*** $p < 0.001$.

In the table in which entrepreneurship has a significant positive impact on the dynamic ability of enterprises (β = 0.453; t = 7.241; $p < 0.001$), the organizational innovation environment (β = 0.571; t = 8.782; $p < 0.001$) and the innovation performance of enterprises (β = 0.322; t = 5.204; $p < 0.001$), dynamic capability of enterprises (β = 0.367; t = 5.298; $p < 0.001$), and organizational innovation environment (β = 0.330; t = 4.254; $p < 0.001$) have a significant positive influence on enterprise innovation performance. Therefore, hypotheses H1-H5 are validated. This is consistent with the research results of Li [61], Peng [62], Zhang and Long [63], Cui et al. [64], Peng et al. [65], and other scholars.

*4.5. Mediating Effect*

This study used the Bootstrap method to test the significance of the mediating effect [61]. First, 5000 Bootstrap samples are drawn from the original data ($n = 426$) by repeated random sampling, and then the model is fitted according to these samples, and 5000 estimates of the mediation effect are generated and saved to form an approximate sampling distribution. The mean path value of the mediation effect and these effect values were sorted by numerical size, and the 2.5th percentile and 97.5th percentile were used to estimate the 95% confidence interval of the mediation effect.

This study involves two mediation paths: P1 and P2. Bayesian grammar programming is used to estimate specific mediation effects. For the judgment of the intermediary effect of a specific intermediary path, follow the following steps. Step 1: judge whether there is an intermediary. The indirect effect is significant, indicating that there is an intermediary. The second step is to judge the type of intermediary. If the direct effect is significant, it is a partial intermediary, otherwise, it is the complete intermediary. If it is a partial intermediary, the proportion of intermediary effect can be calculated according to the formula of direct effect + indirect effect = total effect.

First, the mediating effect test of enterprise dynamic ability in entrepreneurship and enterprise innovation performance.

From Table 5 on the P1 path, the 95% confidence interval of the specific indirect effect is [0.077, 0.233], which does not contain 0, so the mediating effect is significant. The 95% confidence interval for the standardized direct effect was [0.229, 0.493], excluding 0, so some of the mediating effects were meaningful. The specific indirect effect value is 0.146, the total effect value is 0.685, and the mediating effect ratio is 0.146/0.685 = 0.213, that is, when the entrepreneurial spirit on the P1 path has an impact on the innovation performance of enterprises, 21.3% of the variance is caused by the mediating variable enterprise dynamic capability. H6 is verified. This is consistent with the research results of Wei [44] and other scholars.

**Table 5.** Mediating effect.

| Mediating Paths | Indirect Effect | | | Direct Effect | | | Total Effect | | |
|---|---|---|---|---|---|---|---|---|---|
| | SE | 95% CI | *p* | SE | 95% CI | *p* | SE | 95% CI | *p* |
| P1: entrepreneurship→ enterprise dynamic capability → enterprise innovation performance | 0.146 (0.040) | [0.077, 0.233] | 0.001 | 0.330 (0.097) | [0.139, 0.519] | 0.001 | 0.685 (0.056) | [0.572, 0.790] | 0.001 |
| P2: entrepreneurship→ organizational innovative environment → enterprise innovation performance | 0.209 (0.055) | [0.112, 0.326] | 0.001 | 0.330 (0.097) | [0.139, 0.519] | 0.001 | 0.685 (0.056) | [0.572, 0.790] | 0.001 |

Second, the mediating effect test of organizational innovation environment in entrepreneurship and enterprise innovation performance.

From Table 4, on the P2 path, the 95% confidence interval of the specific indirect effect is [0.112, 0.326], which does not contain 0, so the mediating effect is significant. The standardized direct effect is significant, so part of the mediating effect is significant. The specific indirect effect value is 0.209, the total effect value is 0.685, and the mediating effect ratio is 0.209/0.685 = 0.305, that is, when the entrepreneurial spirit on the P2 path has an impact on the innovation performance of enterprises, 30.5% of the variance is caused by the mediating variable caused. H7 is supported. This is consistent with the research results of Xia [46] and other scholars.

The above analysis shows that when entrepreneurship has an impact on corporate innovation performance, 51.8% (0.355/0.685 = 0.518) of the variation is caused by the mediation variables of corporate dynamic capabilities and organizational innovation environment.

Through the above empirical analysis, it is proved that all the hypotheses of this study are tenable.

## 5. Conclusions and Implications

This study mainly discusses the influence mechanism of entrepreneurship on enterprise innovation performance and introduces enterprise dynamic capability and organizational innovation environment as mediating variables. This part summarizes the research conclusions, and illustrates the theoretical contributions and managerial implications of this paper and its research limitations and prospects.

### 5.1. Theoretical Contributions

Through empirical analysis, this study draws the following conclusions.

First, entrepreneurship has a significant positive impact on the dynamic ability of enterprises and the environment of organizational innovation. From this point of view, entrepreneurs need to constantly improve the organizational environment innovatively. Second, enterprise dynamic capability and organizational innovation environment have a positive significant impact on enterprise innovation performance. From this perspective, entrepreneurs need to make efforts to enhance the dynamism of their employees. Third, entrepreneurship has a noticeable positive impact on the enterprise innovation performance. Therefore, entrepreneurs need to rethink their innovative mind and put it into practice. Finally, enterprise dynamic capability and organizational innovation environment play a partial mediating role between entrepreneurship and innovation performance. From this point of view, it seems that the role of entrepreneurs is very important in creating dynamic capabilities and innovative environments for companies.

The above empirical research shows that entrepreneurship has a positive effect on the dynamic ability of enterprises and the organizational innovation environment ($\beta = 0.453$, $t = 7.241$, $p < 0.001$; $\beta = 0.571$, $t = 8.782$, $p < 0.001$), which is consistent with the research results of Li [62], Peng [63], and other scholars. It shows that the stronger the entrepreneurial spirit of the enterprise, the dynamic ability and organizational innovation environment of the enterprise will also be improved.

The above empirical research shows that enterprise dynamic capabilities and organizational innovation environment have a positive role in promoting enterprise innovation performance ($\beta = 0.322$, $t = 5.204$, $p < 0.001$; $\beta = 0.368$, $t = 5.324$, $p < 0.001$), which is consistent with the research results of Zhang and Long [64], Cui et al. [65], and other scholars. It shows that the improvement of enterprises' dynamic capabilities and organizational innovation environment is conducive to enterprises' improvement of innovation performance.

The above empirical research shows that entrepreneurship has a positive role in promoting enterprise innovation performance ($\beta = 0.367$; $t = 5.298$, $p < 0.001$), which is consistent with the research results of scholars such as Peng et al. [66]. It shows that in SMEs, the stronger the entrepreneurial spirit of managers, the corresponding innovation performance will also be improved. If an enterprise wants to improve its innovation performance, it needs managers with a strong entrepreneurial spirit to influence the enterprise.

The above empirical research shows that the hypothesis of the intermediary role of enterprise dynamic capabilities and organizational innovation environment between entrepreneurship and innovation performance has been verified (the intermediary effect ratio is 0.213 and 0.307, respectively), which is consistent with the research results of Wei [44], Xia [46], and other scholars. However, it plays a partial intermediary role, which indicates that there are other factors that play a role in entrepreneurship and innovation performance, such as R & D, organizational learning, tacit knowledge, and so on. In future research, we need to continue in-depth analyses to find more influencing factors.

The research conclusion of this paper provides new research ideas for the theoretical circle. In detail, from the literature review, it is found that scholars' attention was mainly paid to enterprise performance, but less attention was given to innovation performance. Additionally, by summarizing the relevant theories of entrepreneurship, enterprise

dynamic capability, organizational innovation environment, and enterprise innovation performance, it is found that the research on these four concepts is relatively rich, and there are relatively more studies on the relationship between two of the four, but there are few studies on the influence mechanism between entrepreneurship, enterprise dynamic capability, organizational innovation environment, and innovation performance, At present, scholars have not put it into a research conceptual model. Therefore, this paper studied the relationship between entrepreneurship, enterprise dynamic ability, organizational innovation environment, and enterprise innovation performance in small and medium-sized enterprises. It further complements the discussion on the role mechanism of entrepreneurship on enterprise innovation performance. In empirical research, in the aspect of empirical research, the method of questionnaire survey is used to implement the specific implementation, and the design of the questionnaire is carried out on the basis of clear variable connotation to better verify the relationship between entrepreneurship and enterprise innovation performance, and explore the value of entrepreneurship, supplement, and development of entrepreneurship.

*5.2. Management Implications Environment*

In cultivating entrepreneurial spirit aspects, entrepreneurship is an important factor in the development of enterprises. Therefore, leaders with entrepreneurship should be cultivated and promoted. They can contribute through stimulating the entrepreneurial spirit of managers, and then creating a corporate culture that advocates innovation, taking entrepreneurial spirit as the company's institutional system, motivating employees to continue to innovate both materially and spiritually, and advocating full-staff learning, therefore improving corporate innovation performance. In addition, entrepreneurship is a vital resource for SMEs, and managers' entrepreneurship has an important impact on the dynamic ability of the enterprise, the environment of organizational innovation, and the innovation performance of the enterprise. In view of this, managers of Chinese small and medium-sized enterprises should continue to learn and improve their own ability and literacy. They should be brave in innovation and should not hesitate to take advanced actions. They should evaluate the internal and external environment of the enterprise and improve their risk-taking ability. Entrepreneurs should also create an organizational environment to innovate and a corporate environment of continuous learning within the enterprise, continuously improve the dynamic ability of the enterprise, and inspire employees to continue to innovate with cultural environment so as to improve the innovation performance of the enterprise.

Improving the dynamic capabilities of enterprises. The components of the dynamic capability of an enterprise mainly include the ability to learn and absorb, the ability to integrate resources, and the ability to change and reorganize the organization. In terms of learning absorptive capacity, first, create a corporate culture of full-staff learning and lifelong learning, guide employees to continuously learn and absorb new knowledge with an invisible culture, and constantly improve their own abilities; second, establish a sound internal training system, and use the system to supervise employees continuing to carry out self-innovation; third, establish and improve external training and communication channels, provide employees with opportunities where they can go out to exchange and learn and can learn the latest research results from industry-leading companies or research institutes to enrich the knowledge structure of employees. In terms of resource integration capabilities, enterprises should pay attention to the rational allocation of internal and external resources, pay attention to changes in the external environment, grasp industry market trends, continuously integrate resources, and strengthen cooperation and exchanges with external institutions so as to enhance their dynamic capabilities. In terms of enterprise reform and reorganization, attention should be paid to improve the communication and collaboration of various functional departments within the enterprise, stimulate the internal vitality of the enterprise, and the enterprise managers can grant decision-making autonomy to the corresponding departments, teams, or workers in a timely manner.

Creating an environment of organizational innovation. The continuous innovation of an enterprise is related to the development quality of the entire social economy, which affects the efficiency of market resource allocation and also affects the development of the enterprise itself. Therefore, enterprises must create an environment of organizational innovation to promote continuous innovation. First, create a relaxed, active, and innovative corporate culture environment, which will affect employees invisibly. Corporate values should be established to form entrepreneurship, create an organizational environment suitable for the long-term development of entrepreneurship, and to provide a good environment for cultivating corporate innovation capabilities. Second, the advocacy of continuous innovation of the enterprise is formed into an enterprise system to urge employees to innovate; in terms of performance system, a reward mechanism for employee innovation is implemented to motivate employees' innovation from both material and spiritual aspects. Employees should be encouraged to innovate boldly in order to improve the innovation performance of the enterprise, and at the same time make them establish correct risk awareness and give certain support when employees' innovative behavior fails. Third, enterprise managers should attach importance to and participate in innovation, enhance communication between superiors and subordinates, and encourage employees to innovate and trust employees. At the same time, enterprises should actively cooperate with external institutions and research institutes, and jointly invest in research and development to improve their innovation capabilities and innovation performance.

### *5.3. Limitations and Prospects*

### 5.3.1. Research Limitations

In terms of theoretical background, although this study summarizes and studies the related theories background of entrepreneurship, enterprise dynamic ability, organizational innovation environment and enterprise innovation performance, it still needs further discussion on the relevant theoretical system.

In terms of sample size, restricted by the conditions, 426 questionnaires were responded to, and the coverage was low. In order to achieve better representativeness, the number of samples needs to be further expanded.

In the development stage of the enterprise aspects, at different stages of development, there may be differences in various variables. In this study, there is no division of the development stage of the enterprise, which can be enriched in subsequent studies.

### 5.3.2. Research Prospects

According to the limitations of the above study, the research improvements and prospects of this paper are put forward.

In-depth study can be conducted on entrepreneurship, dynamic capabilities of enterprises, organizational innovation environment, and relevant theoretical knowledge of enterprise innovation performance, and provide reliable theoretical support for further research.

Increase the sample size and select representative SMEs from all over the country and conduct a sample survey according to the nature and scale to verify the generality of the conclusions of this study.

In future research, we can define the development stage of enterprises, such as start-up enterprises, developing enterprises, and mature enterprises, and study the relationship between the four variables in different development stages.

**Supplementary Materials:** The following supporting information can be downloaded at: https://www.mdpi.com/article/10.3390/su141912063/s1. File S1. Questionnaire.

**Author Contributions:** Conceptualization, F.C. and J.-h.S.; Supervision, J.-h.S.; Writing—original draft, F.C.; Writing—review & editing, F.C. All authors have read and agreed to the published version of the manuscript.

**Funding:** This research received no external funding.

**Institutional Review Board Statement:** WS-2022-17.

**Informed Consent Statement:** Not applicable.

**Data Availability Statement:** Not applicable.

**Conflicts of Interest:** The authors declare no conflict of interest.

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
