# Peer review of "Impact of Entrepreneurship on Innovation Performance of Chinese SMEs: Focusing on the Mediating Effect of Enterprise Dynamic Capability and Organizational Innovation Environment"

_sustainability, doi:10.3390/su141912063_

Round 1

Reviewer 1 Report (Previous Reviewer 3)

Most of the initial queries have not been addressed

(i) The  work still contains many grammatical blunders

(ii) The research problem is not assertively articulated

(iii) The work is not current as a significant proportion of the citations are very old.

Authors should please give a point-by-point indication of how they addressed the queries for clarity

(iv)  The authors failed to indicate the gaps in literature. What is then, the justification of the study?

(v) the authors failed to indicate what the sample size is and how the sample size was determined. What sampling technique did the authors use?

(vi)             They used a questionnaire to elicit the desired responses from the respondents. What is the source of the questionnaire? How did they administer it? Answers to these questions are critical, not only for a proper understanding of the study, but also for possible replication of study

(vii). the findings are still not properly defined

(viii) The authors failed to discuss the findings of the study. The findings of the study were not related to those of extant studies to provide justifications for deviations or consistencies with previous studies

Author Response

Dear reviewer:

  Best regards,

  FANG CUI, JAEHOON SONG

Reviewer 2 Report (New Reviewer)

Thank you for providing me with the opportunity to review your paper. I have enjoyed reading it. However, I believe that further work is necessary before the paper is suitable for publication.

One major concern relates to the lack of academic strength of the paper. More specifically, it is not very clear what contribution your paper makes to the extant literature.

My review will follow the format taken by the paper.

1. The introduction should present the structure of the paper.

2. The empirical analysis section is at a fairly general level. Some supplementary literature must be added to compare and contrast the key findings with the existing study. The discussion needs greater engagement with the literature to bring it up to an appropriate level.

3. The conclusion has some issues. The conclusion must be based on your results and discussion, but some of the paragraphs presented in the conclusion section belong to the results section. They are just results, for example, “The above empirical research shows, that enterprise dynamic capabilities and organizational innovation atmosphere have a positive role in promoting enterprise innovation performance(β=0.322, t=5.204, p<0.001; β= 0.368, t=5.324, p<0.001), which is consistent with the research results of Zhang and Long [64], Cui et al. [65] and other scholars”. The conclusion must provide the main theoretical and practical contributions and implications of the research, but it does not seem to me that the theoretical contributions presented correspond to real theoretical contributions. That is, although the authors provide a summary of the existing literature in the field, it remains unclear how exactly they extend this literature.

4. Recheck the references and their style are according to the journal requirements, and in-text and end-text should be the same and vice versa.

I hope my feedback on this paper will help the authors to improve their work.

Author Response

Dear reviewer:

  Best regards,

  FANG CUI, JAEHOON SONG

Reviewer 3 Report (New Reviewer)

Referee ReportResearch on the Impact of Entrepreneurship on Innovation Performance of Chinese SMEs”

Major comments: This paper aims to understand the relationship between entrepreneurship, enterprise innovation performance, dynamic enterprise ability and organizational innovation environment. The author(s) form(s) a questionnaire to collect data with a sample size of 426. Author(s) use(s) structural regression model. Results show that entrepreneurship is positively correlated with dynamic ability, organization environment, and firms’ innovation performance. In addition, dynamic capabilities and the organization environment are positively correlated with firms’ innovation performance and play a mediating role between entrepreneurship and innovation of the firm. The paper needs to be restructured.

·       Abstract: The model is missing. Mention that you made and collected the data.

·       First and third findings in the abstract can be written as one

·       Literature: Replace the hypothesis with subtitles in the literature since there is evidence that they are correct from previous literature mentioned in the paper.

·       Mention the citation of the theoretical model in the literature labeled “hypothesis 7”

·       Research design: The sample of 426 firms is widespread in all of China, then the sample is small. If those firms are concentrated in an area, mention the area. In addition, include the questionnaire in an appendix at the end.

·       Mention the model and explain its use in the research design not the results section.

·       Title and inside the text: replace the word “organizational climate” with “organization’s environment”. Climate is linked more with the actual climate and weather.

In my opinion, the paper needs to follow the above comments to be considered for publication

Author Response

Dear reviewer:

  Best regards,

  FANG CUI, JAEHOON SONG

Round 2

Reviewer 1 Report (Previous Reviewer 3)

the authors did not address all the queries posed in the first review. I suggest that they attend to all the queries. For the avoidance of doubt, the unanswered queries are attached 

Author Response

Dear reviewer:

Best regards,

FANG CUI, JAEHOON SONG

Reviewer 2 Report (New Reviewer)

Most of my suggestions have been properly responded. However, some of them are still neglected. All suggestions should be improved before accepting for publication. The references and their style must be in accordance with the requirements of the journal.

Please check and improve all of my comments carefully.

Author Response

Dear reviewer:

Best regards,

FANG CUI, JAEHOON SONG

This manuscript is a resubmission of an earlier submission. The following is a list of the peer review reports and author responses from that submission.

Round 1

Reviewer 1 Report

The article has a great cognitive value. It is recomemnded to continue the research taking into account the sectoral division of enterprises as well the division of enterprises operating both on the domestic and international market.

Author Response

Dear reviewer:

Thank you for your suggestions on my article. The following is my revision. Please review it.

According to the opinions of the review committee, the paper's English language and style were modified. Thank you for your suggestions on my article, which makes my article look clearer and smoother. Thank you very much for your hard work.

Kind regards,

FANG CUI, JAEHOON SONG

Reviewer 2 Report

The manuscript examines the relationship of entrepreneurship, enterprise innovation performance, enterprise dynamic ability and organizational innovation atmosphere of small and medium enterprises in China. A research on 426 companies has been conducted. 

The paper is well structured and clearly written. However, some improvements are necessary:

The information provided about the sample is not sufficient, the authors should explain from which population it was selected, how the contacts were obtained, what representativeness was ensured by the sample (geographically, sectoral, dimensional?), which sectors the companies belong to, how they were contacted, what kind of interview was done for both the paper questionnaires (interviews administred by an interviewer or completed independently?) and electronic online questionnaires (CAWI ?, CATI? autonomous compilation?).

The measurement of innovation performance is particularly important to make the study useful at the practical implications level. How was the innovation performance measured? The authors talk about of 9 measurement indicators that were adopted. It is important to report and comment them, in order to check if they can provide a valid indication on the influence on innovation performance (it is also important to connect this with the literature in order to make clear the concept of innovation performance and how to measure it). Furthermore, in table 1, about enterprise innovation performance, 7 is the number of items reported. Were they 9 or 7?

Conclusion are poor and do not provide significant contribution to the existing theory. They should be expanded and improved. Similarly, practical implications are obvious and don’t reveal anything new stating that 1) within companies  an entrepreneurial spirit should be cultivated, 2) dynamic capabilities should be improved, and 3) an atmosphere of organizational innovation is important to be cultivated.

Reviewer 3 Report

A good research problem and a good attempt but there are so many flaws that need t be remedied. Please see my detailed comments in the attached file

Round 2

Reviewer 2 Report

The authors properly responded to the comments, therefore the manuscript can be published 

Author Response

Dear reviewer:

  I am very glad to hear from you. Thank you for reviewing my paper. You have worked hard.

  Best regards,

  FANG CUI, JAEHOON SONG

Reviewer 3 Report

There is scarcely any evidence that the last queries were answered. Please present a point-by-point response to each of the queries posed in the original submission. Where necessary, present your rebbutals 

Author Response

Dear reviewer:

 Kind regards,

 FANG CUI, JAEHOON SONG

Round 3

Reviewer 3 Report

Some of my queries in the previous review have not been addressed/ They include: 

(i)                 The authors failed to indicate the gaps in literature. What is then, the justification of the study?

(i)                 the authors failed to indicate how the sample size was determined

(ii)               What sampling technique did the authors use?

  1.  What is the source of the questionnaire?

(i)                 The findings of the study were not related to those of extant studies to provide justifications for deviations or consistencies with previous studies

Author Response

(The authors gave the same response as above.)
